# Growth of magnetic nanowires along freely selectable ⟨*hkl*⟩ crystal directions

Y. Tao[1,2] & C.L. Degen[2]

The production of nanowire materials, uniformly oriented along any arbitrarily chosen crystal orientation, is an important, yet unsolved, problem in material science. Here, we present a generalizable solution to this problem. The solution is based on the technique of glancing angle deposition combined with a rapid switching of the deposition direction between crystal symmetry positions. Using iron–cobalt as an example, we showcase the simplicity and capabilities of the process in one-step fabrications of ⟨100⟩, ⟨110⟩, ⟨111⟩, ⟨210⟩, ⟨310⟩, ⟨320⟩, and ⟨321⟩-oriented nanowires, three-dimensional nanowire spirals, core–shell hetero-structures, and axial hybrids. Our results provide a new capability for tailoring the properties of nanowires, and should be generalizable to any material that can be grown as a single-crystal biaxial film.

---

[1] Rowland Institute at Harvard, 100 Edwin H Land Boulevard, Cambridge, MA 02142, USA. [2] Department of Physics, ETH Zurich, Otto Stern Weg 1, 8093 Zurich, Switzerland. Correspondence and requests for materials should be addressed to Y.T. (email: tao@rowland.harvard.edu)

Nanowires with controlled morphology[1, 2], size distribution[3], composition, and crystallinity[4] are enabling devices with novel and enhanced functionalities[5], they are advancing the fields of ultrasensitive chemical sensing[6], ultrasensitive mass[7] and force detection[8], wearable pressure gauges[9], single-photon generation[10], renewable energy generation[11], as well as waste energy recovery from both mechanical[12] and thermal sources[3, 13]. Magnetic nanowires, which are at the focus of this study, find important biological and medical applications in targeted drug delivery[14], remote-controlled cell signaling[15], in vivo micromanipulation[16], and nanoscale magnetic resonance imaging[17, 18].

Crystal orientation is an important attribute of matter. It can have a strong impact on mechanical properties, magnetic anisotropy[19], band structure modulation[20], micromachining etching kinetics, and surface reactivity[21]. Although nanowires can nowadays be grown in a variety of shapes, sizes, and materials, controlling the crystallinity has remained challenging[5]. Single crystallinity is especially important in metallic, magnetic, and semiconducting nanowires, for they typically have a lower defect density. Motivated by these prospects there has been a long-standing effort[2, 4, 22–25] at influencing the single-crystal growth orientation through epitaxial substrate crystal orientation[4], precursor pressure[23, 24], process temperature[24], catalyst particle size[22], and catalyst composition[25]. Most of these approaches are, however, limited to specific crystal directions. Having a technique at hand that could extend growth to an arbitrary design orientation would be highly desirable.

Here, we describe a method for growing single-crystal nanowires along a freely selectable $\langle hkl \rangle$ Miller index. Our method relies on the technique of glancing angle deposition (GLAD)[1, 14, 26–28] combined with a rapid switching of the deposition direction between crystal symmetry positions. Using magnetic iron cobalt as an example material, we demonstrate growth of nanowires along $\langle 100 \rangle$, $\langle 110 \rangle$, $\langle 111 \rangle$, $\langle 210 \rangle$, $\langle 310 \rangle$, $\langle 320 \rangle$, and $\langle 321 \rangle$ crystal directions. We further show that angle-switched GLAD can be conveniently extended to fabricate more complex structures, such as three-dimensional spirals and core–shell or segmented heterostructures. These structures exhibit important functionalities for a diverse range of applications.

## Results

### Crystal orientation control through rotational symmetry

The key to controlling nanowire crystal orientation is a process that can form a biaxial thin film[29] with in-plane rotational symmetry. In a biaxial film, all crystallites exhibit the same crystal orientation. One crystal direction is along the surface normal while the other (in-plane) direction is determined by the direction of evaporation. GLAD realizes biaxial thin film growth by operating a deposition process, such as sputtering or electron-beam evaporation, at a high substrate tilt angle $\alpha > 80°$ (Fig. 1). The high substrate tilt causes ballistic shadowing effects between neighboring seeds and leads to the growth of separated columnar structures with typical diameters of tens to hundreds of nanometers[1]. Single-crystalline growth is enabled through evolutionary selection of the fastest growing crystal plane(s) as material is being added. Crucially, if the unit cell has a rotational symmetry around the surface normal, this symmetry is preserved over the entire film.

In this work, we exploit the rotational symmetry of a biaxial thin film to exert full control over the nanowire crystal orientation. Figure 1c illustrates our concept on a biaxial film with a cubic lattice structure. The transparent blue cube visualizes the $C_3$ rotational symmetry of the film with the [111] axis perpendicular to the substrate and a $\{11\bar{1}\}$ plane facing the

source (Source 1). Under appropriate GLAD conditions, the evaporation of material from the source leads to columnar growth along the [100] axis of the cube. By positioning additional sources (Sources 2 and 3) at relative azimuth angles $\phi = 120°$ and $240°$, the growth can be extended to the [010] and [001] axes. Together, the three evaporation sources therefore allow growth along all three basis vectors of the cubic unit cell.

In the following, we demonstrate that the growth can be extended to arbitrary $\langle hkl \rangle$ crystal directions by a vectorial addition of the growth directions. We achieve vectorial growth by simultaneous evaporation from all three sources while judiciously adjusting the deposition rates. In particular, if $r_1$, $r_2$, and $r_3$ are the deposition rates for each source, we show that the resultant nanowires have principal axis along $[hkl] \propto r_1[100] + r_2[010] + r_3[001]$. To choose a certain orientation, we then simply select deposition rates in proportion to the desired $h$, $k$, and $l$.

**Biaxial FeCo thin film.** Our experimental model material was the iron–cobalt alloy $Fe_{0.65}Co_{0.35}$. This system has the highest reported saturation magnetization at room temperature among all thermodynamically stable alloys ($\mu_0 M_{sat} \sim 2.4$ T, ref. [30]), and is the material of choice for applications where a high magnetic moment is required[14, 16–18, 31]. To initiate our process, we first established the conditions for growing biaxial FeCo films using only Source 1 (see Supplementary Note 1, Supplementary Methods: Section 5.7). Figure 2a and b shows scanning electron micrographs (SEM) of FeCo columns grown at a tilt of $\alpha = 85°$ with a fixed rotation angle $\phi = 0$. The nanowire tips exhibited striking faceting behavior with three mutually perpendicular facets intersecting at a corner oriented normal to the substrate. Energy-dispersive X-ray (EDX) spectroscopy revealed a constant iron–cobalt composition along the length (Supplementary Methods: Section 4.5). Analyses by transmission electron microscopy (TEM, Fig. 2c, Supplementary Methods: Sections 4.1 and 4.4) and transmission Kikuchi diffraction (TDK, Supplementary Methods: Section 4) confirmed that the columns were single crystals with a $\langle 100 \rangle$ direction oriented along the wire axis. The measured lattice constant of $2.89 \pm 0.01$ Å was in good agreement with literature values of $2.87 \pm 0.01$ Å for $Fe_{11}Co_5$ and $Fe_5Co_3$[32]. We further measured the $\beta$ angle to be $55° \pm 2°$, corresponding to the angle of $54.7°$ between $\langle 111 \rangle$ and $\langle 100 \rangle$ axes in the cubic lattice (Supplementary Methods: Section 5.1). Together, our data are consistent with the biaxial texture illustrated in Fig. 1c.

**Arbitrary orientation control.** We next demonstrated the arbitrary orientation control. Rather than evaporating material from three independent sources, we used a single stationary source with a rotatable substrate platen in combination with time sharing. The advantage of time sharing is a much more precise control of the relative deposition rates. The desired $\phi$ direction was adjusted by rotating the platen between preset positions at $0°$, $120°$, and $240°$. If the switching is sufficiently fast compared to the rate of growth, material is vectorially added as $[hkl] \propto t_1 r[100] + t_2 r[010] + t_3 r[001]$, where $t_1$, $t_2$, and $t_3$ are the dwell times and $r$ is the deposition rate of the common source. For our setup, the maximum switching speed led to transit times of $\sim 100$ ms between source positions, allowing for dwell times of $0.5$–$1.5$ s per directions. This corresponded to $1.5$–$4.5$ nm of material being added per dwell.

Figure 3 shows a collection of nanowires grown along $\langle 100 \rangle$, $\langle 110 \rangle$, $\langle 111 \rangle$, and $\langle 321 \rangle$. Further orientations included $\langle 210 \rangle$, $\langle 310 \rangle$, and $\langle 320 \rangle$ (Supplementary Methods: Sections 5.2 and 5.3, Supplementary Table 1, and Supplementary Note 2). The crystal

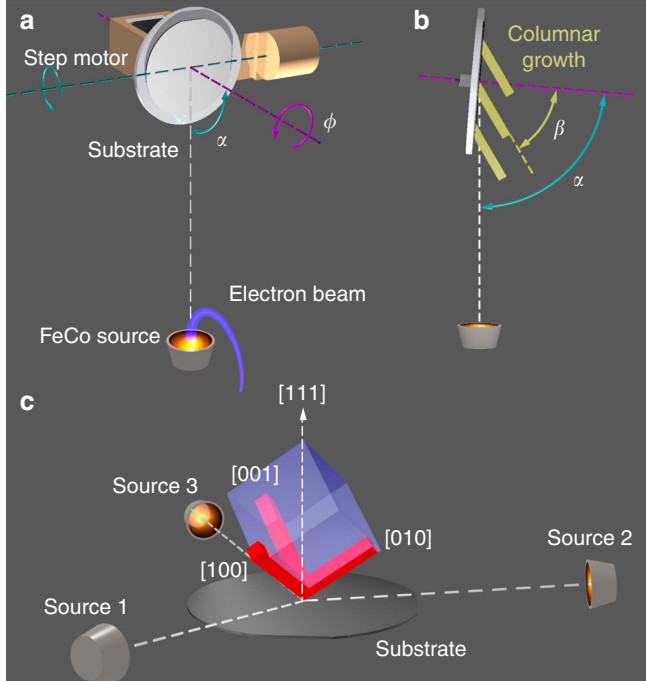

**Fig. 1** Control of nanowire crystal orientation by vectorial glancing angle deposition (GLAD). **a** Schematic of the GLAD setup with definitions for the $\alpha$ and $\phi$ angles. **b** The columnar growth results from the ballistic shadowing effect induced at large substrate tilt angles (here $\alpha = 85°$). The nanowire tilt angle $\beta$ is in general smaller than $\alpha$ and related to the crystal structure. **c** Control of crystal orientation is achieved by vectorial addition of material from several source positions. Illustrated is the growth of a material with cubic symmetry. Three sources located at $\phi = 0°$, $120°$, and $240°$ promote growth along the [100], [010], and [001] basis vectors. In the experiment, a single source is time shared via a rapid switching between $\phi$ positions using a programmable stepper motor

orientation could be directly inferred from the arrangement of 100 facets (Fig. 3i–l) and from the growth angle $\beta$ (Supplementary Methods: Section 5.1). In addition, we have confirmed the crystal orientation and single crystallinity by TDK for several nanowire samples (Fig. 4a). All inspected nanowires showed the expected designed crystal orientations. Our results demonstrate that it is possible to grow FeCo nanowires with an arbitrary $\langle hkl \rangle$ crystal orientation.

**Sculpting three-dimensional structures**. An important feature of GLAD is the possibility to grow more complex structures, especially three-dimensional nanowires[1, 2, 14, 33] and material hybrids[5]. Such higher-level structures have greatly enhanced the application range of nanowire materials[5]. We have explored whether the control over crystal orientation can be extended to such structures. As a first example, Fig. 4 shows a FeCo spiral consisting of 12 wire segments. Such magnetic spirals are being considered for in vivo micromanipulation via remote control fields[14, 16]. The spiral was assembled from alternating $\langle 100 \rangle$, $\langle 110 \rangle$, $\langle 111 \rangle$, and $\langle 210 \rangle$ wire segments (Fig. 4a and Supplementary Methods: Sections 5.3 and 5.4) by intermittently switching the growth orientation during an uninterrupted evaporation run (see Supplementary Methods: Section 2.4 and Supplementary Table 2). The TKD analysis (Fig. 4d) showed that single crystallinity was preserved over the entire spiral structure. Although misoriented crystallites were occasionally present (see arrow in Fig. 4d), the growth was clearly self-correcting and such crystallites were effectively eliminated from the entire structure.

**Multi-material nanowire heterostructures**. The crystal orientation-controlled growth can also be extended to core–shell heterostructures and axial hybrids that are difficult to synthesize with other methods (Fig. 5a). A first example of a material hybrid included a $\langle 100 \rangle$ FeCo nanowire that was conformally coated with a thin film of $Al_2O_3$ using atomic layer deposition (ALD). The alumina coating served to protect the nanowire from corrosion

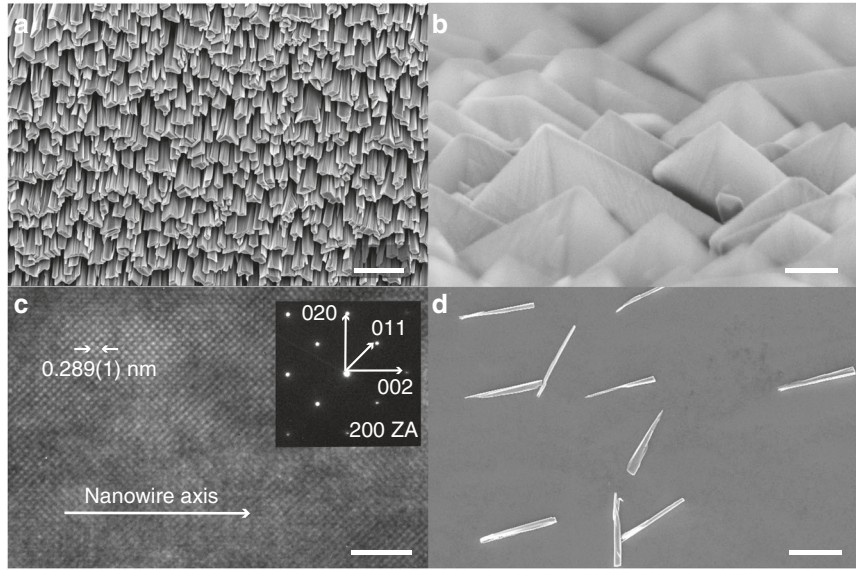

**Fig. 2** Growth of single-crystal FeCo nanowires. **a** Scanning electron micrograph (top-down view) of an as-fabricated sample. Scale bar is 2 μm. **b** Scanning electron micrograph of the same sample as seen from the FeCo source. Scale bar is 200 nm. **c** High-resolution transmission electron micrograph (HRTEM) of a FeCo nanowire viewed down a $\langle 100 \rangle$ zone axis. Inset shows Fourier transform of data collected from a larger region of the same nanowire. Data are consistent with a body-centered cubic (bcc) structured crystal with a $\langle 100 \rangle$ direction coinciding with the nanowire growth axis. Scale bar is 2 nm. **d** Isolated nanowires after transfer to a secondary substrate. Scale bar is 2 μm

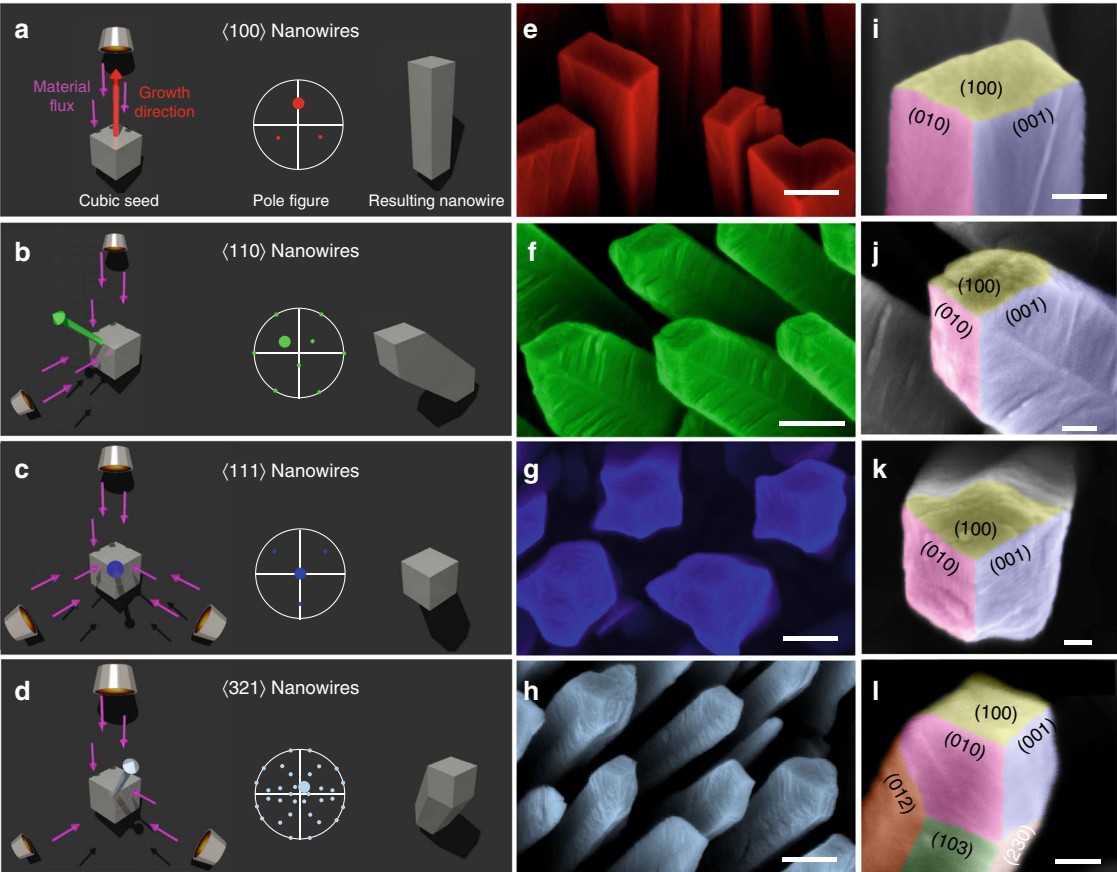

**Fig. 3** Growth of FeCo nanowires along arbitrary ⟨hkl⟩ crystal orientations. **a–d** Schematic of material vapor flux pattern used to generate specific crystal orientations. Sizes of crucibles represent different dwell times: small (0.5 s), medium (1.0 s), and large (1.5 s). Pole figures and resulting nanowire morphologies are also sketched. **e–h** Color-coded SEM micrographs of the corresponding experimental realizations. Scale bars are 200 nm. **i–l** High-resolution SEM micrographs of the nanowire end indicating the crystalline facets. Scale bars are 50 nm. All views are along the substrate normal

and to improve the biocompatibility, both of which are key for biological and medical applications[34]. Using a dynamic cantilever magnetometry measurement[18], we found that the magnetization of the nanowires was high, on the order of $\mu_0 M_{sat} \sim 2.0$ T (Supplementary Methods: Sections 3 and 7), and that as little as 5 nm of $Al_2O_3$ rendered the magnetization completely resistant even to a harsh acidic environment (Fig. 5b, c). The conformal coating was only possible because GLAD produced free-standing nanowires with direct access to all side walls, in contrast to template-based growth techniques[35].

Another class of important heterostructures are stacked multilayers composed of two or more materials along the nanowire axis. Stacked multilayers can be produced by switching the evaporation source in situ during growth. As examples, we have fabricated ⟨100⟩-FeCo nanowires with end segments made from Si, Ho, and Nb. The elemental composition of the heterostructures could be clearly resolved by EDX spectroscopy measurements (Fig. 5d, e). Although these examples mainly served to explore the heterogrowth, we note that such nanowires are expected to support a range of applications. Ho–FeCo and Si–FeCo (Supplementary Methods: Sections 6) nanowires, for example, are ideal probes and actuators for high-resolution magnetic force microscopes[18, 31] and rotating nanoelectromechanical system devices[36], respectively. Nb–FeCo, on the other hand, represents a ferromagnet–superconductor hybrid that could display interesting vortex physics due to the small size of the superconducting film[37]. A further notable heterostructure included a lateral multilayer consisting of two ⟨100⟩-FeCo layers

sandwiching an interlayer of MgO. We found that, thanks to the well-defined end faces of GLAD-grown single-crystal nanowires, very thin (∼ 5 nm) and sharply defined interlayers can be deposited (Fig. 5f and Supplementary Methods: Sections 5.5). Looking forward, the MgO interlayer may be used to tune the exchange, dipolar, and Néel's "orange-peel" couplings[38, 39] to induce an antiparallel arrangement between the two ferromagnetic layers[38]. Such a magnetic configuration would result in a sharp boundary in the magnetization direction and roughly double the gradient of the magnetic stray field, with important applications in ultrasensitive magnetic moment detection[31].

## Discussion

Our experiments demonstrate that single-crystal FeCo nanowires can be grown along arbitrary crystal directions and with a three-dimensional structure customizable by design. Although we have not tested any further materials, we expect that the orientation-controlled growth can be extended to any material that can be deposited as a biaxial film (Supplementary Note 3). In particular, biaxial growth has been demonstrated for metals[29], alloys[40], semiconductors[26], and ceramics[29]. In addition, there is much leeway for further expanding the range of accessible materials and improving crystalline quality and material geometric uniformity, for example by varying substrate temperature, deposition rate, dwell times, $\alpha$ angle modulation, UHV operation, surfactant addition, doping[40], seed patterning of the substrate surface, and combination with epitaxial and catalytic growth modes

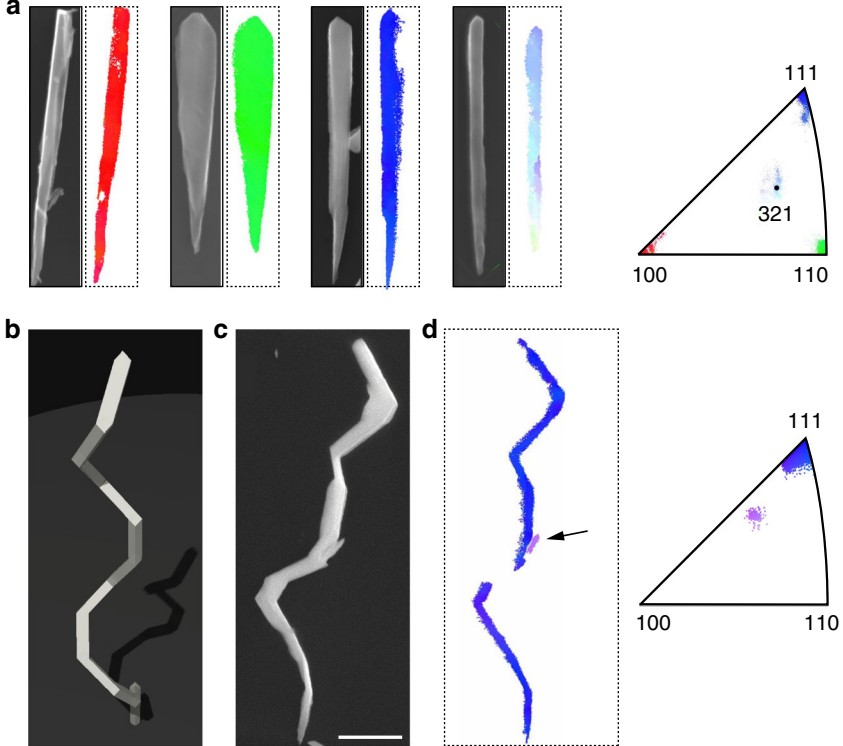

**Fig. 4** Demonstration of three-dimensional growth. **a** Scanning electron micrographs and corresponding transmission Kikuchi diffraction (TKD) patterns for ⟨100⟩, ⟨110⟩, ⟨111⟩, and ⟨321⟩ nanowires that served as the building blocks for 3D growth. Nanowires are 3 μm in length. **b** Schematic, **c** SEM micrograph, and **d** TKD image of a three-dimensional FeCo spiral with 12 segments. Orientation color-coding is with respect to the helical axis. The small segment of missing TKD data is caused by self-shadowing of the spiral with respect to the detector. A small misoriented crystallite is also visible (arrow). Scale bar in **c** is 1 μm

(Supplementary Notes 1–5). Given the unique properties of ⟨hkl⟩-controlled materials, we expect future research to pave the way for many applications requiring tailored, hybrid, and three-dimensionally shaped single-crystal nanowire structures.

## Methods

**Motor control hardware and software**. We constructed a standard GLAD setup by implementing a computer-controlled $\phi$-rotation sample stage with manually adjustable $\alpha$ inside a commercial electron-beam evaporator (Pfeiffer Classic 500) (see figures in Supplementary Methods Section 1.1). Sample $\phi$ rotation was achieved using a step motor (Oriental Motor, model: LCMK245AP). The motor stepping motion (0.1125° step size) was activated by sending a square waveform (10 kHz, 50% duty cycle) generated by an arbitrary waveform generator (HP Agilent Keysight, 33120A) to the motor driver (24 VDC input micro step driver). The stepping direction was chosen via voltage control signals sent to the motor driver from a National Instruments PCI DAQ card. Timing and amplitudes of motor movements were centrally controlled using custom software written in Labview 2009.

**Motor thermal anchoring**. Heat dissipation by convection is absent in a vacuum environment. To achieve efficient thermal anchoring of the step motor, we constructed the motor mount and the $\alpha$ rotation arm from high-purity copper plates (1 cm thickness) and cylinder (4 cm diameter), respectively. The $\alpha$ rotation arm was furthermore heat sunk against a 10-cm-diameter, 30-cm-tall cylindrical copper block in contact with the water-cooled floor of the chamber (see figures in Supplementary Methods Section 1.1).

The efficiency of the thermal anchoring for the motor was experimentally verified by its continuous operation in vacuum for over 1 h. The resulting temperature of the motor was monitored using thermocouple and is plotted in a figure (see figure in Supplementary Methods Section 1.2). In light of these results, all $\phi$-variable depositions were conducted after warming up the motor for 30 min. Given that a typical deposition run lasts ≈1200 s, motor operation heats the sample by no more than a couple of degrees Celsius.

**Evaporation material**. To achieve steady and high deposition rates over tens of minutes, it is crucial to use bulk FeCo (and Co: Supplementary Note 3) materials that had been alloyed by vacuum induction melting (VIM). Otherwise, dissolved

gases and uneven heating could trigger large rate fluctuations, and even sudden explosion of the source material during evaporation. Iron/cobalt starter sources (VIM Fe/Co 65/35 at%, 99.95% pure) machined into the shape of standard evaporation crucibles (29.3 mm top OD × 22 mm bottom OD × 15 mm height) were purchased from Kurt J. Lesker Company (see figures in Supplementary Methods Section 1.3).

The starter source was directly fitted into the cooled copper pocket of the ebeam system without using a liner (see figures in Supplementary Methods Section 1.3). Prior to loading the FeCo source, the copper pocket was meticulously cleaned by polishing with fine sand paper and isopropanol-drenched Kimwipe, until the interior shone like newly machined reddish copper. For reproducible results, we found it important to avoid contamination of the source material by also thoroughly cleaning the sample shutter and other parts of the system, where debris of other evaporated materials could peel off and accidently fall into the source. Starter source was wiped clean with isopropanol-drenched Kimwipe before loading into the ebeam pocket.

Gloves were worn during the entire chamber preparation procedure and changed promptly when contaminated by visible marks.

**Substrates**. Substrates used in this study include wafers of Si(100)/Si(111) (native oxide and Si-H terminated), sapphire, glass, kapton tape, and coatings of Poly (methyl methacrylate) (PMMA) and ZEP on flat Si(100). Native oxide-bearing silicon, sapphire, glass, and Kapton substrates were cleaned in IPA and blown dry with nitrogen. PMMA and ZEP coatings were used after spin coating. Hydrogen-terminated silicon substrates were mounted into the system and pumped down to vacuum within 30 s of etching in 1% hydrofluoric acid (HF).

During the preparation of the source material, gloves were worn at all times while handling and loading sample substrates.

**Chamber vacuum preparation**. Chamber pressure before the start of deposition runs was in the low $10^{-6}$ mbar range. Before each nanowire growth evaporation, the FeCo source was first cleaned by evaporating away some material with the substrate shutter still closed. This step usually led to a slight increase in chamber pressure. Then, the chamber was further pumped by evaporating Ti at a rate of 1 nm/s over 2 min (still with shutter closed). The chamber pressure dropped down into the low $10^{-7}$ mbar range. Immediately after the Ti pumping step, the nanowire growth step was conducted by opening the shutter after FeCo evaporation rate had stabilized to the target value. The electron beam was held fixed with neither

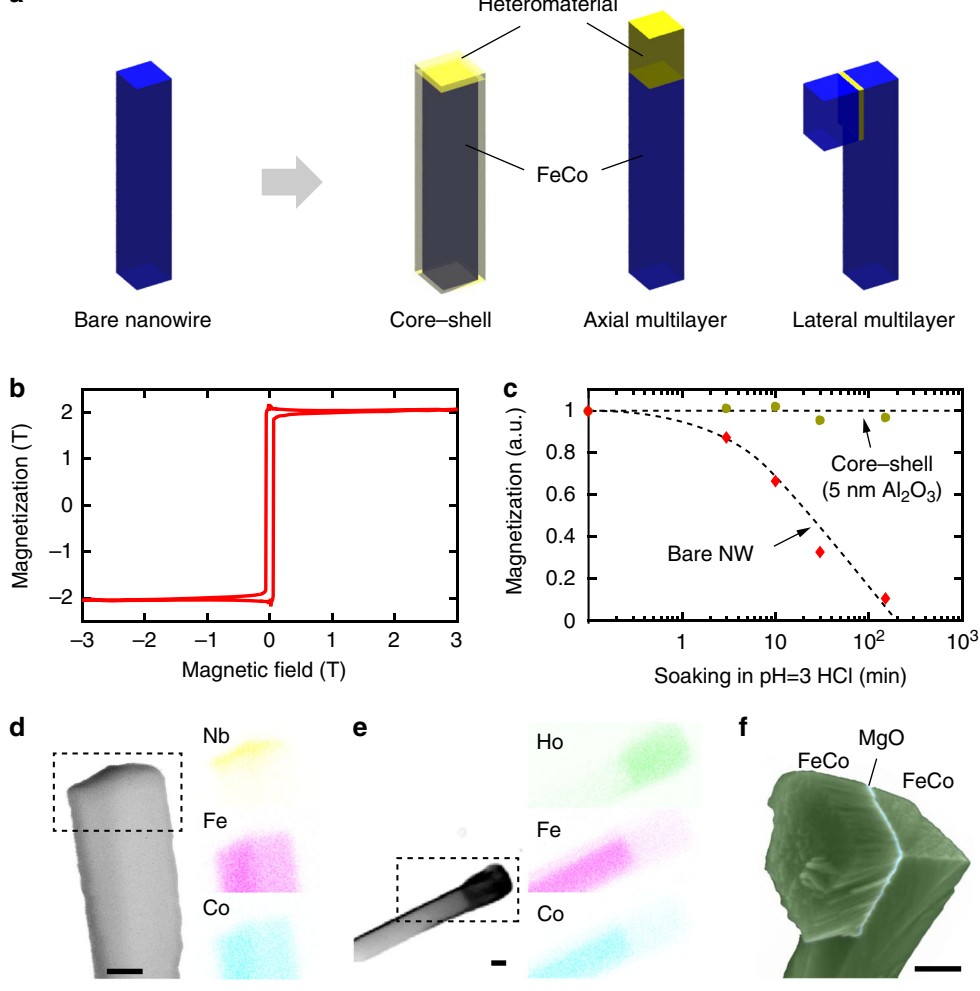

**Fig. 5** Hybrid nanowires. **a** Cartoon illustrating bare, core–shell, axial hybrid, and lateral hybrid nanowires. **b** Magnetization curve of a single ⟨100⟩ FeCo nanowire measured by dynamic cantilever magnetometry[18] along the easy axis. The saturation magnetization $\mu_0 M_{sat} \sim 2.0 \pm 0.2$ T is considerably higher than that reported for other free-standing tips[17]. **c** Magnetization of nanowires following immersion in an acidic medium (pH = 3, HCl). Complete degradation is observed for free FeCo nanowires. Conversely, as little as 5 nm of conformal atomic-layer deposited (ALD) alumina completely protects the magnetic nanowires against degradation. **d**, **e** Secondary electron images and energy-dispersive X-ray spectroscopy images (colored insets) for Nb–FeCo and Ho–FeCo axial hybrids. Scale bars are 100 nm. **f** False-color SEM micrograph of a lateral FeCo hybrid with a 5 nm MgO interlayer. Scale bar is 100 nm

wobbling nor scanning motions to best approximate a point source. Chamber pressure during nanowire growth increased slowly from the $10^{-6}$ mbar range into the low $10^{-5}$ mbar range.

**Nanowire structure fabrication and characterization**. Details of the procedures and parameters used in the fabrication, handling, and characterization of nanowire structures (those presented in the manuscript as well as many additional ones) are provided in Supplementary Methods.

**Data availability**. The authors declare that the data supporting the findings of this study are available within the paper and its Supplementary information files. These data are also available from the corresponding author upon reasonable request.

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

## Acknowledgements

This work was supported by the NCCR QSIT, a competence center funded by the Swiss NSF, Swiss NSF grant 200021_137520/1, ERC Starting Grant 309301, and a Rowland Fellowship. We thank Prof. Rolf Allenspach, Prof. Max G. Lagally, and Prof. Charles T. Campbell for helpful discussions. Christoph Keck, Martin Kloeckner, and Sandro Tiegermann for assistance in sample fabrication. Dr. Fabian Gramm, Dr. Stephan Gerstl, Dr. Karsten Kunze, Dr. Joakim Reuteler, and Dr. Alla Sologubendo at ETH ScopeM for assistance with characterization and helpful discussions.

## Author contributions

Y.T. designed the study, developed the equipment and processes, fabricated the samples, and performed the characterizations. Y.T. and C.L.D. discussed the results and co-wrote the paper.

## Additional information

**Competing interests:** The authors declare no competing financial interests.

