## [Peer Review File · Nature Communications]

Reviewers' comments:

Reviewer #1 (Remarks to the Author):

The authors present a major advance in the fabrication of nanomaterials, with a wide gamut of potential applications, of interest to a wide range of readers. The manuscript is well written; the figures clear and complete; the supplementary material appropriate; and overall an excellent paper, well deserving of publication in Nature Communications.

The analysis methods are well presented, and convincing.

This research solves a long-standing technological problem, in a creative and widely-applicable way. This paper will have a major impact and will form the basis for significant new research in coming years.

I don't see any issues that need to be addressed before publication - please accept as-is.

Reviewer #2 (Remarks to the Author):

The manuscript describes a very nice methodology to fabricate oriented FeCo nanowires and their different variants/structures of FeCo nanowires on different substrates. The results clearly extend the ability of the GLAD technique for crystalline metal nanowire fabrications with controlled growth orientations and complex structures. The demonstrated materials growth control likely will further advance the exploration of the GLAD technique for the synthesis of complex materials. The illustrated concepts will be of interests to the nanowire and materials community. The authors also explore different possible growth mechanisms for the demonstrated methodology. While the discussion is thoughtful, as they point out, more experiments are needed for verifications of the proposed mechanism. In short, the reviewer recommends this manuscript for publication upon minor modifications discussed in the comments below.

Detailed comments:

1. The title of the manuscript is partially misleading. While the authors nicely demonstrated the principle to growth FeCo nanowires with several different orientations, they did not demonstrate the growth of arbitrary orientations. As they pointed out, there is finite time for the substrate to switch between positions. Thus, high index growth orientation could be difficult to achieve. The authors are advised to tone down the claim.
2. The authors claimed that the FeCo nanowires with several demonstrated single orientations are single crystalline. What is the evidence? How many wires were examined? Can they be polycrystalline but with all the assembled crystals having the same orientation? The rough surface of low-index wires (such as [100] orientation in figure 3a) and irregular shape of wires (such as ones in figure 4a) made the reviewer wonder if the single crystalline claim is more complex than shown. Are the composition (Fe:Co) ratios consistent along these wires? Can the authors tell if the lattice strain changes these wires?
3. Shadowing and non-uniform seedings are two limitations of the GLAD technique. What are the percentages of seeds successfully grown into FeCo wires?
4. Nanowires with uniform diameters are useful in many applications. The reviewer is curious on what can be done to improve the uniformity of the demonstrated FeCo wires and what the authors have achieved so far.
5. The authors have nicely demonstrated the fabrication of different heterostructured nanowires. However, the details for their fabrication is missing in the supplementary information. The authors should either report the detailed fabrication procedures for these materials or remove them from the manuscript. It does not serve the materials community by reporting partial results.

Reviewer #3 (Remarks to the Author):

This manuscript describes the synthesis of FeCo nanostructures with specific crystal orientations and geometries through the use of glancing angle deposition (GLAD). By rotating a (111) growth substrate inside an evaporator such that the [100], [010], and [001] crystallographic directions are aligned to the evaporation source for varying periods of times, the authors show that nanowires with rationally chosen crystallographic directions can be synthesized. In addition, they show that spiral structures can be synthesized that are single-crystalline (at high growth rates) and maintain the same crystal orientation throughout the growth process. GLAD is a relatively well established technique, and rather complex, even spiral, structures have previously been demonstrated (e.g. see "Designing nanostructures by glancing angle deposition," Proc. SPIE 5219, Nanotubes and Nanowires, 59 (2003): DOI:10.1117/12.505253). The advancement and novelty of this manuscript in regard to the GLAD technique seems to be the demonstration of single-crystalline structures instead of the more typical polycrystalline materials synthesized by GLAD. In addition, the authors discuss how the magnetic nature of these single-crystalline materials could be advantageous for a number of applications. They show data on the magnetization of the material and demonstrate that heterostructures can be synthesized. The material growth and characterization all appears to be well done. My primary criticism regards the novelty of the technique, considering that GLAD is a rather well-known method, it is not clear that the demonstration of the single-crystalline, direction-controlled growth through epitaxy is really a surprising advancement. If the authors added a unique technological demonstration with these new nanostructures, it might add a bit more to the novelty that seems to be required for publication in Nature Communications. A few minor comments:

1. The authors discuss multiple methods of nanowire growth control in the introduction. They demonstrate control of crystallographic direction, but can they control other geometrical parameters such as nanowire diameter, length, and density? What is the highest aspect ratio they can achieve?
2. In Figure 5d, 5e, and 5f, the authors appear to show SEM images of their materials. Have the backgrounds been cropped from the images? In particular, the false coloring in 5f indicates a thin MgO, as mentioned in the text, but has this thin layer been confirmed by EDS analysis?

Response to referees' comments

Reviewer #1 (Remarks to the Author):

The authors present a major advance in the fabrication of nanomaterials, with a wide gamut of potential applications, of interest to a wide range of readers. The manuscript is well written; the figures clear and complete; the supplementary material appropriate; and overall an excellent paper, well deserving of publication in Nature Communications.

The analysis methods are well presented, and convincing.

This research solves a long-standing technological problem, in a creative and widely-applicable way. This paper will have a major impact and will form the basis for significant new research in coming years.

I don't see any issues that need to be addressed before publication - please accept as-is

We thank the reviewer for finding the work a major advance in the fabrication of nanomaterials and for recommending its publication.

Reviewer #2 (Remarks to the Author):

The manuscript describes a very nice methodology to fabricate oriented FeCo nanowires and their different variants/structures of FeCo nanowires on different substrates. The results clearly extend the ability of the GLAD technique for crystalline metal nanowire fabrications with controlled growth orientations and complex structures. The demonstrated materials growth control likely will further advance the exploration of the GLAD technique for the synthesis of complex materials. The illustrated concepts will be of interests to the nanowire and materials community. The authors also explore different possible growth mechanisms for the demonstrated methodology. While the discussion is thoughtful, as they point out, more experiments are needed for verifications of the proposed mechanism. In short, the reviewer recommends this manuscript for publication upon minor modifications discussed in the comments below.

We thank the reviewer for a thoughtfully positive reading of our manuscript and for recommending its publication. We also thank the reviewer for a number of excellent questions and suggestions that have helped to significantly improve the clarity and completeness of the narrative.

Detailed comments:

1. The title of the manuscript is partially misleading. While the authors nicely demonstrated the principle to growth FeCo nanowires with several different orientations, they did not demonstrate the growth of

arbitrary orientations. As they pointed out, there is finite time for the substrate to switch between positions. Thus, high index growth orientation could be difficult to achieve. The authors are advised to tone down the claim.

The reviewer points out that finite rotation speed of the particular step motor used in our demonstration experiments poses a technical limitation that currently makes it difficult to achieve ultrahigh-index orientations, such as a $\langle 100,100,1 \rangle$ nanowire. This is, however, a technical limitation, solvable with engineering, which we don't believe detracts from the conceptual advance of the current demonstration. (The current motor and deposition conditions are compatible with a maximum of about $\langle 14, 14, 1 \rangle$. See SI Section 9)

We have modified the manuscript to point the reader to a new section (SI Section 9: *Engineering for Ultrahigh-Index Nanowires*) in the supplementary material that contains more detailed analysis and specific prescriptions for future technical improvements. In the new section, we provide four completely independent prescriptions for accessing high-index nanowires. These are 1) Fast step motor, 2) Sample rotation-beam shutter synchronization, 3) UHV system, and 4) Multiple evaporation sources. The analysis shows that ultrahigh-index nanowires (up to $\langle 125000, 1, 1 \rangle$) are in principle possible if motor switching were the only limitation.

In summary, with the help of the reviewer's comment, we have significantly expanded the supplementary material to point interested researchers to several engineering directions, should the need for producing high-index nanowire arise in their research. We continue to believe that the title is appropriate and faithfully describes the results and contents of the work.

2. The authors claimed that the FeCo nanowires with several demonstrated single orientations are single crystalline. What is the evidence? How many wires were examined? Can they be polycrystalline but with all the assembled crystals having the same orientation? The rough surface of low-index wires (such as [100] orientation in figure 3a) and irregular shape of wires (such as ones in figure 4a) made the reviewer wonder if the single crystalline claim is more complex than shown. Are the composition (Fe:Co) ratios consistent along these wires? Can the authors tell if the lattice strain changes these wires?

The reviewer raises important questions about details of the crystallinity and composition of the fabricated nanowires. We have added additional data and discussion in the supplementary information (Sections 4.4 and 4.5) to address these questions.

We first address the issue of the visibly rough surface (Fig. 3a). Based on SEM micrographs, these nanowires have surface roughness on the order of 1nm-10nm. It is indeed important to check whether such roughness could be due to an underlying polycrystallinity. We prepared a new figure in the SI (Fig. S7) showing HRTEM images of the edges and corner of a $\langle 100 \rangle$ nanowire. The images show surface roughness and features ~ 2 -10nm, in agreement with SEM measurement. However, the data also establishes that the rough features are part of the same underlying single crystal.

Closer examination of these images provide important additional insights. First, the FeCo single crystals seem to be covered by a thin layer of crystalline surface layer (roughly 2-3nm in thickness) that is most likely an oxide layer. This sample is more than two years old at the time of HRTEM inspection and the result highlights the stability of the nanowires toward air oxidation. Second, the surface roughness features seem to be bounded by crystalline microfacets. Future study of these features and their evolution in time (ideally *in situ* during growth) may help elucidate the mechanism of homoepitaxy.

Next, we address the questions individually:

- What is the evidence? How many wires were examined?

The crystal structure and crystal orientations of the FeCo nanowires were characterized by high-resolution transmission electron microscopy (HRTEM) and by transmission Kikuchi diffraction (TKD). HRTEM images furnish unequivocal evidence of local single-crystallinity by providing atomic-resolution images of portions of the nanowires tens of nanometers in size. Three nanowires were examined by HRTEM and were found to be single-crystalline within fields of view 30nm-50nm. Diffraction pattern collected from a 200nm area shows a single set of peaks.

TKD provides complementary, global, and population-level information by enabling mapping over large sample areas containing multiple nanowires. More than twenty different nanowires and spirals were examined by TKD during the course of this study. All examined wires showed the expected, design crystal orientation over the entire length of the nanowires; the local crystal orientations were consistent, pixel to pixel for 5nmx5nm to 20nm-20nm pixels, over the entire lengths.

- Can they be polycrystalline but with all the assembled crystals having the same orientation? The rough surface of low-index wires (such as [100] orientation in figure 3a) and irregular shape of wires (such as ones in figure 4a) made the reviewer wonder if the single crystalline claim is more complex than shown.

In raising this possibility, the reviewer acknowledges that the HRTEM and TKD data unequivocally establish that the bulk of the material within the nanowires, even if polycrystalline, are aligned in the same orientation. The reviewer suggests that such (hypothetical) domains, instead of forming a continuous lattice spanning the entire nanowire, could be distinct crystallites separated from each other by, presumably, intervening crystalline layers, amorphous layers, or grain boundaries.

We first consider the possibility where the intervening 'glue' were misoriented crystallites. TKD can detect crystalline domains on the order of 2 nm. In none of the TKD maps we acquired have we observed hints for lines or planes of misoriented crystallites separating the majority, uniformly-oriented crystalline domains. Therefore, if the nanowires were indeed oriented polycrystals, boundary between them can only be nearly sub-nm in thickness. In this size regime, which approaches the lattice constant of ~0.3nm, the difference between nanocrystalline and amorphous is blurred and we might as well consider the intervening 'glue' to be amorphous.

There is no *a priori* conceptual reason why such amorphous interlayers could not exist. These boundary planes would be either perpendicular or parallel to the nanowire axis. If perpendicular barriers existed, it would however be difficult to explain how new crystal domains on top of an amorphous barrier would want to replicate the orientation of the crystallite underneath the boundary; homoepitaxial growth would be expected to stop in the absence of direct physical contact with a crystalline substrate.

Amorphous barriers parallel to the nanowire might be compatible with epitaxial growth along the entire nanowire. Figure 4a shows that the bottom end of the nanowires have a taper, terminating in a sharp end from which they nucleate; the nanowires increase in thickness initially as they win the initial competitive growth phase. Due to the small size of the 'feet', which is most likely a single-crystal domain (See new SI data: Fig. S12), we would expect the hypothetical amorphous boundaries, should they form, to form during elongation, after nucleation. However, there is no clear mechanism or driving force for this to occur.

The data and physical intuition therefore suggest that grain boundaries and other types of crystallographic defects (edge dislocation, stacking faults, and twin boundaries) as the most probable defects (rather than an amorphous phase or nano-polycrystalline phases) due to the highly uniformly oriented nature of the (hypothetical) domains: it would therefore be scientifically more accurate to speak of single-crystals (with lattice imperfections).

- Are the composition (Fe:Co) ratios consistent along these wires?

Based on EDX data collected during the course of this study, the composition ratio (Fe:Co) is constant along FeCo nanowires at the resolution and sensitivity of the technique. The ratio agrees with specification of Kurt Lesker for the evaporation starter sources used. In addition to the data in Fig. 5d-e, we have added another example of EDX mapping of a longer segment of a nanowire in the SI (Fig. S8).

This result is intuitive: the ebeam evaporation process used vacuum induction-molten starter sources, which, due to good heat transfer through the bulk metal, only melts locally during evaporation where the electron beam strikes. In the absence of beam scanning (to better approximate a point source to minimize angular dispersion of the vapor), an area about 3mmx3mm melts. During a run, or several runs with the ebeam focused on the same spot, we have noticed the 3mmx3mm hole to grow deeper, indicating that fresh material is being continuously replenished as the liquid component evaporates away. Together, local melting of the source and continuous replenishment of the liquid puddle eliminates distillation effect and keep the material composition of the source vapor constant. We further established through measurements described in Section 8 of the SI that the temperature of the substrate and nanowires vary little and stay very close to room temperature during the evaporation process, leading to expectedly high and constant sticking coefficients.

- Can the authors tell if the lattice strain changes these wires?

Strain in crystalline structures invariably arise in the presence of defects. The FeCo nanowires produced in this study certainly contain crystalline defects of various types. For example, we discussed the possibility of dislocations and grain boundaries in b) above. There could also be point defects that would result when foreign atoms are incorporated into the crystal, such as oxygen, nitrogen, and carbon that invariably permeate a high vacuum system operating at 10^{-6} torr.

While we have not undertaken a study of the strain in the nanowires, the contrast variation we see in the HRTEM images could be a potential hint for the existence of strain (Fig. S7). Further studies are needed to verify this possibility and to quantify its magnitude.

3. Shadowing and non-uniform seedings are two limitations of the GLAD technique. What are the percentages of seeds successfully grown into FeCo wires?

Non-uniform seeding on unpatterned substrates is indeed a limitation of the GLAD technique. GLAD can be performed on intentionally seeded substrate surfaces, however. In the case of amorphous pillars, 100% pillar yield with respect to starting seeds is obtained (10.1088/0957-4484/16/9/052; 10.1021/jp1060528). The situation is obviously different in non-seeded GLAD and in GLAD of single crystal wires.

The current study has focused on developing single-crystal orientation control on un-patterned substrates. As a result, stochastic inter-nanowire competition lead many starting seeds to lose the race and stop growing at an early stage. This selection process is in fact critical for the formation of a biaxial film. To provide an illustration of the process, we have included an additional SEM image showing the cross sectional view of the spiral sample (Figure S12). It is clear that the initial, spontaneously nucleated seeds are tiny (~ 10 nm) and that a significant percentage of them fail to grow into mature nanowires. A rough estimate from looking at the SEM image suggests that on the order or 10% or less of the initial seeds successfully grow into FeCo wires. Future research aiming to increase the survival rate of rationally designed and placed seeds should consider, among others, substrate-directed epitaxy, or seeds with properly orientation placed at strategic locations on the substrate.

4. Nanowires with uniform diameters are useful in many applications. The reviewer is curious on what can be done to improve the uniformity of the demonstrated FeCo wires and what the authors have achieved so far.

It is also clear that there is a gradual increase of nanowire diameter during growth. We have also discussed how diameter uniformity and nanowire mean diameter may be better controlled in responding to comments by Reviewer #3.

Developing methods to control nanowire diameter distribution within the context of crystal orientation-controlled GLAD is indeed useful in many applications of nanowire materials. Among existing methods for nanowire synthesis, templated growth (as electrodeposition into anodic alumina nanopore arrays)

and catalyzed growth (AuNP-catalyzed Si VSL) are particularly adept at controlling nanowire diameter. In the realm of growth by glancing angle deposition, similar ideas have been successfully explored and applied.

While we have not initiated further study of the subject, there are several avenues for future work in this area. The different avenues we describe here are included as a new section in the supplementary material to guide interested researchers (SI Section 11.2).

Diameter control is a multi-faceted task involving intra-wire uniformity, inter-wire uniformity, and the average diameter.

- 1) Within the realm of amorphous and polycrystalline pillars, intra-wire uniformity can be controllable by phi-angle modulation techniques (10.1088/0957-4484/16/9/052, 10.1088/0957-4484/15/7/018, 10.1002/9781118847510: Chapter 3) techniques to decrease the effect of stochastic fluctuation in self-shadowing areas. Since the control of crystal orientation also relies on phi-angle modulation, the extent to which this approach can be applied to single-crystal GLAD remains to be explored. It is, however, entirely possible that substrate swing with small phi-angle excursions (10.1002/9781118847510, Figure 3.16 c) can be compatible with single-crystalline control that uses much larger phi excursions.

Within the realm of single-crystal pillars, a recognized mechanism for intra-wire diameter variation is gradual increase of wire diameter with length due to adatom diffusion across the Ehrlich-Schwoebel (ES) barrier (10.1103/PhysRevLett.101.266102). Therefore, adjusting the available thermal energy with respect to the ES barrier heights (diffusion down single or multiple steps have different activation energies) can be used as a handle to tune the steady-state nanowire diameter during growth (10.1103/PhysRevLett.101.266102, Fig. 3). It is conceivable that dynamically tuning the substrate temperature during a deposition run could be used to modulate intra-wire diameter (10.1021/jp1060528: Figure 3 and Figure 4).

- 2) Inter-wire uniformity and mean pillar diameter can be controlled by lithographic seeding (doi: 10.1116/1.2737436, Figure 8) or by the effective diffusion length of the adatoms, adjustable by substrate temperature and by evaporation rate (10.1103/PhysRevLett.101.266102; 10.1038/srep16826).

This and further discussions on geometric control can be also found in our response to the first comment by Reviewer #3 (SI Section 11).

5. The authors have nicely demonstrated the fabrication of different heterostructured nanowires. However, the details for their fabrication is missing in the supplementary information. The authors should either report the detailed fabrication procedures for these materials or remove them from the manuscript. It does not serve the materials community by reporting partial results.

We have added details of the fabrication procedures for these heterostructure nanowires in the SI (Section 2.5).

Reviewer #3 (Remarks to the Author):

This manuscript describes the synthesis of FeCo nanostructures with specific crystal orientations and geometries through the use of glancing angle deposition (GLAD). By rotating a (111) growth substrate inside an evaporator such that the [100], [010], and [001] crystallographic directions are aligned to the evaporation source for varying periods of times, the authors show that nanowires with rationally chosen crystallographic directions can be synthesized. In addition, they show that spiral structures can be synthesized that are single-crystalline (at high growth rates) and maintain the same crystal orientation throughout the growth process. GLAD is a relatively well established technique, and rather complex, even spiral, structures have previously been demonstrated (e.g. see "Designing nanostructures by glancing angle deposition," Proc. SPIE 5219, Nanotubes and Nanowires, 59 (2003): DOI:10.1117/12.505253). The advancement and novelty of this manuscript in regard to the GLAD technique seems to be the demonstration of single-crystalline structures instead of the more typical polycrystalline materials synthesized by GLAD. In addition, the authors discuss how the magnetic nature of these single-crystalline materials could be advantageous for a number of applications. They show data on the magnetization of the material and demonstrate that heterostructures can be synthesized. The material growth and characterization all appears to be well done. My primary criticism regards the novelty of the technique, considering that GLAD is a rather well-known method, it is not clear that the demonstration of the single-crystalline, direction-controlled growth through epitaxy is really a surprising advancement. If the authors added a unique technological demonstration with these new nanostructures, it might add a bit more to the novelty that seems to be required for publication in Nature Communications. A few minor comments:

We thank the reviewer for a careful reading of our manuscript and for providing constructive criticism that helped us improve the manuscript.

Reviewer is correct in pointing out that GLAD is a well-known method, with its unique set of advantages. Unfortunately, the lack of crystallinity control has prevented this patently convenient method from being competitive with many other methods for nanomaterial synthesis, such as MBE, VLS growth, and hydrothermal methods, etc, that can more reliably afford single-crystalline products. This is because high-quality, single-crystalline material have become increasingly mandatory in cutting-edge research and technological applications; figures of merits most often improve by orders of magnitude going from amorphousness, to polycrystallinity, to single-crystallinity.

In the present manuscript, not only is it established that GLAD can provide high-quality nanomaterial that no other methods can produce, it has also become the *only* method that can, at present, exert *full* control over nanowire crystal orientation and shape.

We believe this advancement to be major and are confident that it will result in a commensurate impact on material science and technology.

1. The authors discuss multiple methods of nanowire growth control in the introduction. They demonstrate control of crystallographic direction, but can they control other geometrical parameters such as nanowire diameter, length, and density? What is the highest aspect ratio they can achieve?

The reviewer mentions several parameters of interest to researchers applying GLAD for producing advanced materials. Controlling them can be crucially important both for subsequent on-chip batch device processing (most notably: density and placement), or off-chip, applications of single-nanowires (most important: diameter and length). Methods to control these parameters have been developed in existing GLAD literature, and their implementations in combination with the new crystal orientation control capability are conceptually often straightforward, but would require substantial additional research work. The main aim of our study was to demonstrate the feasibility of arbitrary $\langle h,k,l \rangle$ crystal growth.

We believe that this contribution of crystal orientation control will spark renewed interest in the field to develop GLAD methods that would lead to more complete control of nanowire parameters, both in geometry and in crystallinity. We have written an additional section in the SI (Avenues for Future Research in Single-Crystalline GLAD.) with the approaches that could be considered for this purpose.

- 1) The density and placement of GLAD pillars can be completely controlled through the use of a two-step electron beam lithography seeding method (10.1063/1.3222911). Despite the rather preliminary nature of this report, in which imperfect technical execution (such as over-exposure in the process step in Fig. 1 d-e, evident in data in Fig 2 b) of the proposed concept led to broadening of the pillars in the design locations, optimization of fabrication procedure would enable arbitrary and simultaneous control over density, placement, and crystal orientation.
- 2) Diameter control is more challenging and requires intra-wire uniformity, inter-wire uniformity, and the average diameter:
 - For amorphous and polycrystalline pillars, intra-wire uniformity can be controllable by phi-angle modulation techniques (10.1088/0957-4484/16/9/052, 10.1088/0957-4484/15/7/018, 10.1002/9781118847510: Chapter 3) techniques to decrease the effect of stochastic fluctuation in self-shadowing areas. Since the control of crystal orientation also relies on phi-angle modulation, the extent to which this approach can be applied to single-crystal GLAD remains to be explored. It is, however, entirely possible that substrate swing with small phi-angle excursions (10.1002/9781118847510, Figure 3.16 c) can be compatible with single-crystalline growth.
 - For single-crystal pillars, a recognized mechanism for intra-wire diameter variation is gradual increase of wire diameter with length due to adatom diffusion across the Ehrlich-Schwoebel (ES)

barrier (10.1103/PhysRevLett.101.266102). Therefore, adjusting the available thermal energy with respect to the ES barrier heights (diffusion down single or multiple steps have different activation energies) can be used as a handle to tune the steady-state nanowire diameter during growth (10.1103/PhysRevLett.101.266102, Fig. 3). It is conceivable that dynamically tuning the substrate temperature during a deposition run could be used to modulate intra-wire diameter (10.1021/jp1060528: Figure 3 and Figure 4).

- Inter-wire uniformity and mean pillar diameter can be controlled by lithographic seeding (doi: 10.1116/1.2737436, Figure 8) or by the effective diffusion length of the adatoms, adjustable by substrate temperature and by evaporation rate (10.1103/PhysRevLett.101.266102; 10.1038/srep16826).

3) Nanowire length can be precisely controlled by adjusting the duration of the evaporation run. Alternatively, different growth lengths can be achieved in the sample evaporation run by placing substrates at different distances from the source, as the rate of growth decreases quadratically with source-sample separation.

4) Maximum-achievable aspect ratio is limited only by engineering details of the evaporation chamber. Two factors contribute to this parameter, corresponding to the maximum length and the minimum diameter:

- The duration of the evaporation run establishes the maximum nanowire length. In our setup, the maximum is limited by the fact that there is a single quartz crystal rate monitor inside the chamber. This quartz crystal balance fails after a period of material deposition, and opening the chamber to exchange to a new rate monitor effectively terminates the deposition run. Ebeam evaporator chambers with multiple quartz crystal microbalance exist and would allow longer nanowires to be grown. Eventually, the amount of evaporation material in the source may also become limiting. Modern commercial evaporators can host > 8 hearths and would enable wires with length above 1 cm to be grown, provided, of course, that a sufficient number of quartz crystals monitors could be operated in tandem and/or exchanged *in situ* via specially designed load-lock

- The aspect ratio can also be changed by diameter control, as discussed under 2) above.

While the current study involved producing a nanowire structure with maximum aspect ratio of about 30 (spiral in Fig. 4d), increasing the length and decreasing the diameter, as discussed above, would enable substantially increasing this parameter by at least an order of magnitude.

2. In Figure 5d, 5e, and 5f, the authors appear to show SEM images of their materials. Have the backgrounds been cropped from the images? In particular, the false coloring in 5f indicates a thin MgO, as mentioned in the text, but has this thin layer been confirmed by EDS analysis?

Figure 5d and 5e (left sides) are data taken with ETD on samples dispersed on TEM grids with a 10nm of carbon film, so there was no appreciable background (due to negligible back-scattering and secondary electron generation within the amorphous carbon with respect to within dense metal nanowires).

5f is a SEM image with background removed for clarity. False-coloring faithfully respects the original contrast in the raw SEM data. While we have not confirmed the chemical composition of this thin layer by EDS analysis, the sharp transitions in material composition/structure is evident based on SEM. We have prepared an additional supplementary figure (Fig. S13) to highlight the clear material contrast in the original, raw-data form.

Summary of changes

- 1) We have added a new section in the SI to address engineering of ultrahigh-index nanowires (Section 9) and have added a note in the manuscript to point readers to it.
- 2) We have added additional data (Fig. S7) and a new subsection in the SI to show that the nanowires are single-crystalline despite surface roughness revealed by SEM inspection.
- 3) We have added additional data (Fig. S8) to show that the Fe:Co ratio holds constant within the resolution and sensitivity of EDX.
- 4) We have added additional data (Fig S12) to show the cross-sectional view of the substrate-FeCo interface. These data illustrate the process of competitive growth during the initial stages of biaxial film selection and nanowire formation.
- 5) Fabrication procedures for hybrid nanowires and structures have been added to the SI (Section 2.5)
- 6) We have added a new section in the SI (Section 11) to discuss avenues and strategies to combine crystallinity control with controls over other geometric parameters relevant to nanowires and their devices.
- 7) Raw SEM data for FeCo-MgO-FeCo nanowires have been added to the SI (Fig. S13).

Reviewers' Comments:

Reviewer #2:

Remarks to the Author:

The authors provide direct answers to most of Reviewer #2's questions.

Reviewer #2 recognizes the excellent intellectual merits of the work and the authors' ability to control the growth directions of nanowires with high precision. However, Reviewer #2 is not convinced that the authors have demonstrated that they could fabricate nanowires with "arbitrary" $\langle hkl \rangle$ crystal directions. The inductive theoretical engineering argument is sketchy because there is a finite time for the evaporated atoms to reach the substrate and the sticking coeff. of the atoms on the growing nanowires are not one. So there is a limit for the orientation control. Having an infinitely faster stepper motor may solve one of the many limitations to achieve "arbitrary control". One can make a theoretical argument for the existence of infinitely precise machine, but reality often does not fit all the assumptions. So a demonstration is desirable to make this significant claim.

In short, Reviewer#2 supports the publication of the manuscript. Whether the title fits the claim as it is is left for the editor to evaluate.

Reviewer #3:

Remarks to the Author:

After reviewing the revised manuscript and response to all reviewer comments, it appears that the authors have done a good job at addressing concerns raised in the initial reviews, and the manuscript is suitable for publication.

REVIEWERS' COMMENTS:

Reviewer #2 (Remarks to the Author):

The authors provide direct answers to most of Reviewer #2's questions.

Reviewer #2 recognizes the excellent intellectual merits of the work and the authors' ability to control the growth directions of nanowires with high precision. However, Reviewer #2 is not convinced that the authors have demonstrated that they could fabricate nanowires with "arbitrary" $\langle hkl \rangle$ crystal directions. The inductive theoretical engineering argument is sketchy because there is a finite time for the evaporated atoms to reach the substrate and the sticking coeff. of the atoms on the growing nanowires are not one. So there is a limit for the orientation control. Having an infinitely faster stepper motor may solve one of the many limitations to achieve "arbitrary control". One can make a theoretical argument for the existence of infinitely precise machine, but reality often does not fit all the assumptions. So a demonstration is desirable to make this significant claim.

In short, Reviewer#2 supports the publication of the manuscript. Whether the title fits the claim as it is left for the editor to evaluate.

We thank Reviewer #2 for finding our answers to previously raised questions direct, for recognizing the excellent intellectual merits of the work, and for supporting the publication of the manuscript.

The final remaining concern of Reviewer #2 is in the choice of the word "arbitrary" in the title. After careful consideration, we recognize the reviewer's concern and believe "freely selectable" to be a better descriptor of this experimental work. This is because the word "arbitrary" may bring with it the connotation of theoretical and mathematical absoluteness that is indeed difficult to achieve with any real-world experimental system. In contrast, real-world nanowires would be better described as $\langle h, k \pm \delta k, l \pm \delta l \rangle$, where δk and δl are experimental uncertainties. For this reason, the title now reads: *Growth of magnetic nanowires along freely selectable $\langle h k l \rangle$ crystal orientations.*

For completeness, we analyzed the two processes suggested by the Reviewer to see how they would contribute to uncertainties in crystal orientation and to estimate the magnitude of the uncertainties: 1) "finite time for the evaporated atoms to reach the substrate" and 2) "the sticking coeff. of the atoms on the growing nanowires are not one."

For each of the two suggested causes, we first analyze the process based on an assumption of a continuous stream of matter. We found that if matter were infinitely divisible to provide continuous vapor streams, statistical variations do not arise, neither from the finite vapor travel time nor from a none-unity sticking coefficient. We then discuss why it is the discreteness of atoms that leads to variations δk and δl on the order of 0.01%-0.1% through a statistical fluctuation in the rate of deposition. This discussion is provided in the Supplementary Material as section 12. As a result of this analysis and the conceptual advanced detailed in this study, we believe that regardless of the existence of small directional uncertainty, "freely selectable" is a most appropriate replacement of "arbitrary" in the title.

Reviewer #3 (Remarks to the Author):

After reviewing the revised manuscript and response to all reviewer comments, it appears that the authors have done a good job at addressing concerns raised in the initial reviews, and the manuscript is suitable for publication.

We thank Reviewer #3 for supporting the publication of the manuscript.

Summary of changes

- 1) We have changed the title of the manuscript to *Growth of magnetic nanowires along freely selectable $\langle h k l \rangle$ crystal orientations*.
- 2) We have added discussion of the intrinsic limitation in the precision of orientation control as Supplementary Material section 12.